# Older Adults’ Perceptions toward Walking: A Qualitative Study Using a Social-Ecological Model

**DOI:** 10.3390/ijerph18147686

**Published:** 2021-07-20

**Authors:** Ka-Man Leung, Kai-Ling Ou, Pak-Kwong Chung, Cecilie Thøgersen-Ntoumani

**Affiliations:** 1Department of Health and Physical Education, Education University of Hong Kong, Hong Kong, China; leungkaman@eduhk.hk; 2Department of Sport, Physical Education and Health, Hong Kong Baptist University, Hong Kong, China; pkchung@hkbu.edu.hk; 3Department of Sports Science and Clinical Biomechanics, University of Southern Denmark, 5000 Odense, Denmark; cthogersen@health.sdu.dk

**Keywords:** active aging, physical activity, Hong Kong, walking, thematic coding

## Abstract

Objectives: In this study, we aimed to investigate older adults’ perceptions of their walking experiences, using the social-ecological model as a guiding framework and to propose future walking intervention content. Methods: Thirty-eight participants (19 women; 47% from private elderly centers; mean age = 72.8 (SD = 7.4 years) took part in semi-structured interviews. Qualitative data analysis software QSR-NVivo was used for thematic coding. Results: Thematic deductive analysis revealed pertinent themes at the individual level (health benefits and barriers, fall risk, perseverance, and walking as a suitable activity for older adults), social environment level (social support and social interaction), physical environment level (density, land-use mix, and connectivity; perceived safety, pedestrian facilities (benches, quality of walking paths and sidewalks, and aesthetics), other pedestrian behaviors, and weather, and policy level (lack of walking programs in the community, and supportive culture for an active lifestyle). Discussion: Our findings provide insights for the planning of future multilevel walking intervention programs for older adults in Hong Kong. It is suggested that future walking intervention should include professionals (e.g., physiotherapist or coach) in a group setting, practical walking recommendations such as proper walking posture, and additional fun activities for older adults.

## 1. Introduction

Population aging is a growing challenge in many countries. By 2043, approximately one in three people in Hong Kong will be aged ≥ 65 years. The United Nations [1] predicts that, by 2050, Hong Kong will have the world’s fifth largest proportion increase of older adults. The increase in aging will lead to a rise in the old age dependency ratio from 218/1000 (218 nonworking people supported by 1000 workers) in 2016 to 357/1000 in 2026, which indicates the prospect of a dwindling labor force in the future [2]. The increase in the average age and reduction of the labor force will increase the dependency ratio and burden of medical and welfare services in society.

Although the benefits of physical activity (PA) are well documented [3,4], approximately half of older adults in Hong Kong aged 60–69 years are inactive (i.e., they engage in less than 30 min of moderate- or higher-intensity PA in a week). Less than 30% perform sufficient PA (i.e., at least 150 min of moderate- or higher-intensity PA in a week) to obtain health benefits [5]. Therefore, the Hong Kong government has emphasized the importance of promoting active aging and creating age-friendly environments to mitigate these challenges [6].

Walking is a type of PA that is affordable and carries a low risk of injury to older adults. It can also be easily assimilated into daily life and increases the likelihood of adopting a healthy lifestyle [7]. Walking has numerous health benefits that are critical to older adults, such as improved executive function [8], improved mental health [9], greater self-rated health, improved mobility and independence in daily life, reduced risk of falling, and slowed age-related decline in functional capacity [10,11]. Consequently, considering the benefits of walking to older adults’ health, understanding older adults’ perceptions toward walking and developing walking interventions to promote healthy aging are important.

Although several robust quantitative reviews [12,13,14] and other studies [15,16] have investigated PA and walking in older adults, only a few studies in this field have used qualitative approaches [17]. Some qualitative studies investigated perceptions of walking in middle-aged and older adults in Canada [18], Taiwan [19], Australia [20,21], and the United States [22]. Two qualitative studies examined the environmental factors influencing older adults’ walking habits in Belgium [23] and the United Kingdom [24]. For instance, Grant and colleagues [18] interviewed 31 informants at neighborhood and municipal levels in four different neighborhoods in Ottawa, Canada, to investigate the perception of creating socio-political walkable neighborhoods. The results suggested that improving walkability in a community requires the development of policies and guideline for implementation. Lee and colleagues in Taiwan [19] examined the perception of brisk walking among middle-age and older adults. The results found brisk walking was a good way to promote and maintain middle-aged and older adults’ health, and build social relationship and social interactions. Two studies were conducted in Australia [20,21] and both of these studies suggested to make group-based walking programs appealing and feasible to a diverse group of residents. Leading with walking ambassadors was also a recommended strategy to promote walking in older adults. In the United States [22], Marquez and colleagues investigated older adults’ perceived factors related to walking in their neighborhood, and found individual and environmental factors were main influences in older adults’ walking and PA.

Most of these qualitative studies were conducted in Western countries with lower population densities. Evidence suggested that residential density had positive associations of walkability [13]. This can be explained by the fact that people living in high-density areas are more likely to form the habit of transportation walking. However, this inference needs more research to support it, especially in higher population density cities such as Hong Kong [23]. Crucially, only a few of the related studies were based on a solid theoretical framework such as Self-Determination Theory [20,21]. Reeves et al. [24] highlighted the importance of a theoretical foundation in qualitative studies and noted that the use of theories guided data collection (by establishing a theory-based interview guide) and interpretation and helped clarify the underlying causes or influences of the observed phenomena.

Different from other theoretical frameworks such as Self-Determination Theory, which focus on individual and social environments [20], the number of studies applying the social-ecological model [25] in PA-related studies has increased in the past decade. This model addresses more than just individual and social factors, including environmental factors, and assumes multiple interactive influences (e.g., individual, social, and physical–environmental) on a behavior, and these influences interact across levels. The inclusion of environment- and policy-level determinants distinguishes ecological models from other health behavior theories, which emphasize individual determinants [26]. *Individual factors* are the individual attributes affecting an individual’s walking behavior, such as biological (e.g., age and sex) and psychosocial (e.g., self-efficacy) attributes. For example, older adults with higher self-efficacy have stronger walking intentions [27]. The *physical environment* may include land-use mix, parks and open spaces, and perceptions of personal safety [14]. Evidence has suggested positive associations of total walking for transport purposes with residential density or urbanization, walkability, street connectivity, overall access to destinations or services, land-use mix, pedestrian-friendly features, and access to several types of destinations [13]. The *social environment* refers to social support, social cohesion, and social capital, including culture. Leung and Chung [28] conducted a cross-sectional study and examined the associations between the physical environment, social environment, and walking in 450 older adults in Hong Kong. The results revealed that positive physical environment facilitators and social environments promoted walking. *Policy* factors include policies regarding the development of institutions and regulations for improving health and access to health centers, such as reducing medical costs and increasing PA resources for older adults at both the local and national levels. The social-ecological model has been used to investigate PA [29] and walking [28,30,31] in older adults. The social-ecological model enhances our understanding of the dynamic, interrelated, and multifaceted relationships between people and their environments and guides the development of more comprehensive interventions for health promotion.

Therefore, in order to fill the gaps in the literature, the aim of this study was to address the aforementioned research gaps by using the social-ecological model [32] as a guiding framework to examine Hong Kong older adults’ perceptions of existing walking experiences, as well as older adults’ suggestions for future walking intervention. A qualitative approach enables the collection of in-depth information and provides a more comprehensive understanding of how different factors promote and impede walking habits in older adults. A deeper understanding of older adults’ experiences may support the development of theory-based interventions to promote walking in this demographic.

## 2. Materials and Methods

### 2.1. Participants

This study was conducted between February and June 2019. All the research activities were reviewed and approved by the university’s Institutional Review Board (FRG2/17–18/009). Before the interviews, we sent a letter with details of the research (e.g., participation criteria, aims, and procedures) to elderly centers in Hong Kong. With the consent of the person in charge of private and public elderly centers, we provided recruitment information regarding the study at the monthly meeting of these centers. Advertisements of our study’s recruitment was done through centers’ news board, centers’ newsletter, and personal network of centers’ personnel involved. Members of the centers who were interested in participating in our study then completed a study enrollment form. All eligible participants were further contacted by the person in charge of the centers to determine the details of their interview (time and venue). Again, before the interview, participants were given a study consent form and informed of the details of the study, such as the procedure, potential risks, benefits, confidentiality, compensation, and their right to suspend the interview and withdraw from the study. Participants agreed to join our study by signing the consent forms. For those who enrolled in our study, they all attended our invited interview.

In total, 38 older adults from Hong Kong elderly centers participated in our study. In Hong Kong, the elderly centers such as District Elderly Community Centres (DECC) and Neighbourhood Elderly Centres (NEC) serve adults aged 60 years and above (Social Welfare Department, 2018) and they provide a range of comprehensive services such as educational or sport programs to enable older adults to remain in the community at neighborhood level. This sample size met the sample size guidelines suggested by Moser and Korstjens that no new analytical information arises anymore [33]. Additionally, this sample size took reference of previous related studies [18,19,20,21,22,23,24] that the range of sample size was 15–57. The inclusion criteria for this study were as follows: (1) being aged ≥ 60 years, (2) having no diagnosed cognitive impairment, and (3) being able to walk with or without assistance. According to the United Nations [1], older adults are defined as people aged 60 and above. The Timed Up and Go Test [34] was used to assess participants’ walking competence and the abbreviated mental test [35] was used to assess participants’ cognitive function. Participants requiring more than 20 s for the Timed Up and Go Test and scoring less than 6 in the abbreviated mental test were excluded from the study. The screening tests (i.e., the Timed Up and Go test and abbreviated mental test) were done by the principal investigator (K.M.L.) and a trained research assistant in a meeting room at our partner centers. All participants (*n* = 38) passed the screening tests.

In order to have a “more detailed and balanced picture of the situation (i.e., Hong Kong older adults’ perceptions of existing walking experiences)” [36] (p. 117), older adults were recruited using purposive sampling and were stratified by sex, type of elderly centers (private vs. public), and whether they met the PA recommendation. The PA recommendation refers to the definition adopted in the Healthy Exercise for All Campaign [37]: “Active adults and older adults are those who accumulate at least 150 min of moderate- or higher-intensity PA in a week (or 75 min of vigorous intensity PA or other combinations of various intensities; for example, accumulating 90 min and 30 min of moderate- and vigorous-intensity PA, respectively” [5]. Other than DECC and NEC, which are funded by our government (i.e., public), other elderly centers including self-financed elderly centers were categorized as private elderly centers in this study. Hong Kong is comprised of three main regions geographically. Initially, we targeted to recruit participants from three private and public centers (one from each region in Hong Kong) using convenience sampling. Among the six targeted centers, four private and public centers showed their interest to join our study (one public center from New Terries, three private and public centers were from New Terries and Kowloon in Hong Kong).

In total, we conducted 38 semistructured interviews. All 38 participants were from four elderly centers (three private centers and one public center). The participants (*n* = 38; 19 women; 47% from private centers; 66% with an active PA level) had a mean age of 72.8 (standard deviation = 7.4) years. None of them had previously joined a walking intervention or program and all of them agreed that there was a need to develop walking intervention for older adults. Participant characteristics are summarized in Table 1.

### 2.2. Semistructured Interviews

Semistructured individual interviews were used in this study to understand in-depth the perceptions of older adults’ walking from their perspective. Semistructured interviews involved an interviewer employing a series of predetermined, open-ended questions while remaining focused on the research question [38], which enabled participants to further explore themes or responses. This is frequently used when conducting qualitative research and in health care-related studies [39,40] to collect information. In this study, interviewees shared (1) their beliefs and values related to walking, (2) their perceptions about their current walking behaviors, (3) factors that impacted their walking habits, and (4) their suggestions regarding future walking intervention programs. Interviews began with a standard set of questions (based on the literature) on the walking influences from a socioecological perspective [41,42]. The interviewers attempted to guide the flow of the interview and uncover interviewees’ views. A pilot study was conducted to test a set of interview questions with five older adults and one investigator (P.K.C.) involved in the study (see Appendix A for sample questions). The interviewers adopted a flexible approach and, when necessary, altered the question sequence and/or asked probing questions to facilitate further in-depth conversations. This approach yielded a comprehensive description of the participants’ experiences. Our participants were invited to describe their experiences with little interference from the researcher in terms of guiding or leading the discussion; this enabled us to understand their personal experience with walking, the social and environmental factors that affect their walking, and the improvements necessary for policies and existing or future intervention programs in the community. This method assisted interviewees in describing more comprehensive and complex phenomena by employing diffused thinking; moreover, this approach enabled interviewers to collect walking-related information in individual, social, and environmental levels based on a social-ecological model and immediately provide feedback to interviewees, which maximized information collection within a limited interview time [43].

### 2.3. Procedure

The interviews were conducted by the principal investigator (K.M.L.) and a trained research assistant in a meeting room at our partner centers. The research assistant was a woman who was knowledgeable about PA and older adults, and skilled in qualitative research, including interviewing and data analysis. All interviews were audiotaped for further data analysis, and the interviewers also recorded key information (e.g., personal data and notes) during the interviews by using field notes. The interviewees had no prior relationship with the interviewers. No repeat interview was conducted. The interview locale was quiet and undisturbed, and only the interviewers and interviewees were present during the interview. After the interview, all interviewees received a HK$100 supermarket voucher in recognition of their contribution to the study. The interviews lasted for an average of 34 min (range = 16–79 min).

### 2.4. Data Analysis

All interviews were transcribed verbatim and verified alongside the interviewer notes. To facilitate data analysis, qualitative data analysis software QSR-NVivo (QSR, Burlington, MA, USA) was used for thematic coding. This type of analysis is a highly flexible approach that is useful for summarizing key characteristics of large data sets because it compels researchers to take a well-structured approach to data processing that helps produce a clear and organized final report [44]. During coding, data were organized into conceptual categories or themes (e.g., individual and environmental levels) according to the social-ecological model. Two independent coders (K.L.O. and K.M.L.) read, reread, and coded a portion of the interview transcripts independently (i.e., data triangulation) to ensure that coding was performed in accordance with the research questions and to ensure accuracy. Subsequently, the coded transcripts were compared and discussed among the coder and principal investigator again. Coding interview transcripts is an iterative process, and discussions among the researchers facilitated the development of thematic codes that complemented the qualitative data. Any problems in coding were reviewed and finalized by K.M.L. The 32 comprehensive criteria for reporting qualitative research [45] were used to report the results.

## 3. Results

### 3.1. Old Adults’ Perceptions of Their Walking Experiences: Individual Level

#### 3.1.1. Health Benefits and Barriers

Health was one of the topics discussed by the participants. Participants regarded improved health as a motivation for walking and, at the same time, their health conditions as a major factor discouraging walking. Walking enhanced participants’ psychological well-being because it led to happiness and relaxation. A female participant (PRI10) said, “Frequent walking releases endorphins, right? And dopamine. All these give me happiness.” A female participant (PRI5) remarked, “Walking is the easiest way to become healthy. Except (if) you keep thinking about negative stuff while you are walking; otherwise, you are relaxed and happy.” A female participant (PRI18) noted, “The walk itself is the most enjoyable part. I don’t know how to describe. I am happy when I walk and chat with friends.” Therefore, one source of the happiness came from social bonding, as they could chat with their friends while walking.

Another female participant (PRI18) stated, “Walking removes the strain on my knees” and “Walking lowers my blood pressure level.” Other participants stated that walking is “useful for our digestive system, respiratory system … you will move your arms and legs. It is useful to train up the flexibility of your arms and legs” (PRI2) and that walking enhances “movement in joints and relieves their tightness” (PRI13).

Enhanced sleep quality was another health benefit mentioned by the participants. A male participant (PR7) said, “I get sweaty and take a bath. Then I can sleep well” after walking, especially in summer. Another female participant (PRI16) expressed a similar notion that walking enables her to “sleep more and better on the following day.”

However, the participants’ health status sometimes limited their walking habits. Participants highlighted their diseases, such as high blood pressure, Meniere disease, leg pain, and eye disease, as major limitations discouraging them from walking.

“Since I have high blood pressure, walking will be negatively affected if my blood pressure is not stable. If my blood pressure is high, I will notice my speedy breathing just a few minutes after I start walking. Then I need to find a place for rest soon.” (PRI17, a male participant).

Another male participant (PRI5) also noted, “I have not had surgery for my cataract … my perception of distance is worse than before. So, now I walk a lot slower.”

Physical limitations at an individual level may lower some participants’ confidence and interest at interacting (at social environment) with others while walking in a group. “My feet don’t have the strength to take the load (while walking). (Interviewer: Would you join a walking group?) I won’t join because my feet are not good, I walk very slowly, people have to accommodate me, some of my companions will scold me, and I know my level.” (PUB13, female).

#### 3.1.2. Fall Risk

More than one-third of the participants mentioned the risk of falling while walking. They generally understood the seriousness of falls and were concerned about their fall risk factors. This theme, in fact, interacts with physical environment level that participants may increase their fear of falling when they are exposed to bad weather and walking conditions. Some of the interviewees with fall experiences refrained from walking in rainy days.

“You may suddenly fall down if you walk too fast, but it won’t happen if you walk slowly … the road is not good now. It’s easy to (hit something with your foot) and fall down when the road is uneven, and the floor tiles are (also) bad” (PUB6), a female participant concerned about her fall risk.

One male participant (PRI9) reported, “One time it rained and I slipped when I walked over a manhole cover. (I would) not dare to walk while raining.” Another female participant with a fall experience said, “(The) doctor told me not to fall down. Since I have a steel (implant), I will have to undergo surgery if I fall down.”

#### 3.1.3. Perseverance

Some participants stated they had a daily walking habit. They mentioned determination and perseverance in walking despite difficulties encountered while walking, such as rain. In addition, their perseverance sometimes came from their friends’ encouragement.

A male participant: “It is a matter of determination … if you do not have determination, of course you won’t go (walk). It is so easy to give yourself some excuses. If you are determined, even (if) it is raining, you will bring an umbrella out, right? If you do not have determination, it is better for you to stay in bed and rest, right?” (PUB6).

A female participant (PUB1) exclaimed, “No, we still walk when it rains. We walked even when there was a furious storm last time.” Therefore, perseverance is linked to the social environment level. A female participant (PUB17) highlighted this linkage “Mostly (it) is my perseverance. The most important thing is peer support (which encourages me).”

#### 3.1.4. A Suitable Activity for Older Adults

The interviewees considered walking an appropriate PA for older adults because of its safety, flexibility, and low cost. For instance, older adults might walk according to their competence and needs. A female participant (PUB2) reported, “You don’t need to have (a) high level of strength when you’re walking; you can walk as far as you can. When you’re tired, you can take a seat … I walk based on my ability.”

A male participant: “Walking is a type of sport. It’s safe and (you) don’t need any equipment. You can walk with any clothes and sport shoes. You may walk without company. You can walk alone. Especially in Hong Kong, there is lots (of) space, (so) that you may walk anywhere” (PRI7).

### 3.2. Old Adults’ Perceptionsof Their Walking Experiences: Social Environment

#### Social Support and Social Interaction

Participants reflected on the importance of social support from their family doctor, family, and friends, and how such support encouraged them to walk. A female participant (PUB2) mentioned, “When I see the doctor, he tells me I would feel better if I walk more.” Another participant (PRI11) reported that both family and friends encourage her to walk or walk with her. She said, “They (family and friends) told me to walk slowly to prevent a fall … they will come also.” Notably, this social support was reciprocated among friends and participants. PRI17 noted, “I affect them. I didn’t die even I got cancer. I (my health) became better by walking more”.

Participants also shared that they enjoyed walking (at individual level) because of social interaction at group walking. They regarded walking as a joyful activity because they often socialized with people when they walked. A female participant (PRI10) said, “You could do exercises while making friends. Hence, you will become more sociable and happier”. PRI1 noted, “If I walk with other people and we chat, then the time flies. (It feels more) natural and relaxing to walk (while socializing)”.

Generally, most of the participants preferred to walk in a group and valued the higher perceived safety and enjoyment of interacting with others while walking in a group. For instance, two female participants said: “I prefer walking in a group if I go to a place far away, but just because of safety” (PRI12) and “If you walk alone and fall, you have nobody to help you. If you’re with friends, they would hold your hand. It’s like if I walk with you, you would hold my hand when I fall down” (PUB2).

Again, echoing the above individual level, physical limitations can be one of their barriers to joining a walking group. Few expressed worries or inconveniences when they walked in a group, such as a slower walking pace. Some opted to walk alone because they lacked someone to accompany them. A female participant (PRI11) said, “I walk slowly with a stick so no one will wait for me.” Another female participant (PUB3) noted, “Unlike others, I walk so slowly. If they want to chat with me, they have to wait for me. So, they don’t like to walk with me”.

### 3.3. Old Adults’ Perceptions of Their Walking Experiences: Physical Environment

#### 3.3.1. Density, Land-Use Mix, and Connectivity

The analysis of the interviews revealed density, land-use mix, and street connectivity as critical subthemes. Most of the participants avoided walking in busy streets filled with many people (i.e., high neighborhood density). For example, female participants (PRI8 and PUB1) stated: “You cannot walk in (the) tourist area because there are so many people, and I seldom go there” and “… when you go to the pier in the holidays, then it’s crazy. It’s really crowded; it’s really very crowded. There are a lot of people coming in”.

The availability of various shops and services as well as favorable connectivity to their neighborhood encouraged more walking. A male participant (PRI2) said, “My living area is convenient, like (I) can go (to) two 7-Eleven stores or markets or other functional places easily.” Interviewer: “Like walking from one spot to another spot, do you have many routes to choose from?”. PRI2: “Yes, there are plenty”. However, older adults did not welcome pedestrian bridges connecting different areas. A female participant (PRI1) highlighted the challenge that many older adults face: “Older elders do not like it (bridges) (because) you need to go up the stairs when you go to the bridge. Going down the bridge is another big problem … lots (of) elders are not able to (walk down the) stairs.”

#### 3.3.2. Perceived Safety

Perceived safety was a major factor when participants considered where to walk. For instance, participants reported that high traffic volume discouraged them from walking. “The road is filled with cars and dust. When I enter the road to Laguna City, there are many cars and smoke near the factories … so I usually avoid that way (area)” (PRI1). Another male participant (PUB8) further stated, “There is only one road where I live but there are still many cars … (The roads have no safety features) that separate people and cars”. The type of vehicles on their walking path also affected their perceived safety. The same participant (PUB8) revealed, “There is a big car park behind our hill and all kinds of cars will stop here, like huge construction vehicles, dump trucks, (and) trucks with water … that may be dangerous (for those walking nearby)”.

In addition to concerns related to high-volume traffic, participants disliked speeding cars on the road. A female participant (PUB2) stated, “Car speed in Sai Kung (a place) is fast usually. Some older adults cannot see them turning clearly. If the vehicles move too fast, accidents may easily occur”. Therefore, the participants suggested installing more traffic lights to regulate driver behavior. Participants specifically emphasized crossing time on the street.

A female participant: “It’s not clear … There are not many traffic lights … I mean (the time) from green light to red light (goes) too fast … Sometimes people can’t finish crossing the road during the green light. We don’t have such problems because our legs are fine. But (for) people who use sticks or have problems with their feet, the light changes too fast before they can completely cross the road” (PUB1).

#### 3.3.3. Pedestrian Facilities

The participants also mentioned the value of appropriate pedestrian facilities (e.g., benches and sufficient light) as well as the quality of walking paths or sidewalks.

##### Bench

The availability of benches was regarded as a facilitator, enhancing older adults’ walking behaviors. One female participant (PRI8) appreciated such facilities, noting “(if) there are seats in the park, we will take a seat if we are tired”. However, in areas other than parks, the same participants complained about the absence of seats when they walk. Another female participant (PUB2) had a similar opinion: “There are no chairs for us to sit (on). Old adults of course wish to take a seat while waiting for the bus”.

##### Quality of Walking Paths or Sidewalks

Most of the participants were also concerned about the quality of walking paths or sidewalks when they walked. The interviewees regarded walking paths and sidewalks that were wide and flat with sufficient light as high quality. A female participant (PRI1) stated, “I would walk more if the road was easier to walk on”. Interviewer: “What do you mean by easy to walk on?”. PRI1: “A wider road with fewer cars”. Other participants had similar complaints. PRI17 said, “The (shop) owners will put many things outside their shop area … It really affects us. Especially our mainland friends (who) open their luggage in the center of the road. It blocks our way”.

Uneven walking paths or sidewalks also discouraged the participants from walking, due to fear of falling (an individual level factor). A female participant (PUB11) remarked, “There are some uneven ceramic tiles on the road, (and) you don’t know it’s there until you (hit) it. We have complained many times because (once) an old lady tripped (and fell) on the road”. When questioned regarding the ideal walking environment, a female participant (PUB13) replied, “It’s better (if it is a) flat (surface). You know, I’m afraid of falling down while walking. I will be in big trouble if (I) fall down. So, I prefer to walk on a flat road”. Another female participant (PRI10) echoed “The roads must be flat without any pebbles. The pebbles are not flat, and lighting is very important … The sky is dark after sunset; it is difficult to see things clearly. I (have previously) fallen down because I could not see the road clearly. Some of the stairs are not identical in shape; some are higher, some are lower … If you can’t see the road clearly, you adjust your posture, (and) then you may fall easily”.

##### Aesthetics

Many participants indicated that natural environments (e.g., fresh air, flowers, plants, and greenery), cleanliness, and site attractiveness made walking enjoyable, which is linked with psychological benefits such as enjoyment at the individual level. When asking participants why they enjoyed walking, a male participant (PRI4) answered, “It should be the surrounding, flowers, and plants … (for example), the road is prettier than before”. Another female participant (PUB18) stated, “The most enjoyable thing is having my chest up and to breathe in some fresh air. I can also sweat.”

Although they typically avoided walking in crowded areas, participants liked walking to sites that they regarded as attractive.

“The promenades are most suitable for me to walk. I like walking on the promenades the most. Many people park their boats there. There are many things to see; for example, people sell fish there. It’s near the pier; you can go there and see. There are a lot of people (during) the weekends” (PUB2; female).

The participants had some complaints concerning dogs. PUB3, a female participant, said, “Just sometimes, dogs are everywhere.” PUB8 remarked, “They (people) walk their dog and they don’t (act responsibly) as they don’t clean up the dogs’ excrement”. These factors made walking unattractive.

#### 3.3.4. Other Pedestrian Behaviors

Other pedestrian behaviors, such as people walking their dogs, smoking, gathering in large groups (gangs), and exhibiting drunken behavior affected participants’ walking. A male participant (PUB4) stated, “The older adults walk so slow … you know huge dogs may bite sometimes”. Interviewer: “Even (if it’s restrained by its) master, it doesn’t help?” … “Yes, the dog can recognize its master but not strangers, right? People next to the dog will still (be) afraid … (fearing the dog thinks they are thieves). Yes, it is terrible.” Another female participant (PUB18) shared a similar experience: “At nighttime, there are many people and dogs. Therefore, I usually go to the court. Sometimes people smoke”. A female participant (PRI10): “Also, I hate when people smoke in front of me. I am sensitive to cigarettes. When they smoke in front of me, I might take in those smoke. I hate that”.

#### 3.3.5. Weather

Weather was both a barrier and facilitator of the older adults’ walking. Interacting with individual level, walking on hot or rainy days may increase their perceptions of fall risk such as being afraid of falling and getting injured. A male participant (PRI2) said, “What I enjoy most must be the beautiful weather … What I enjoy least is rainy days, so if (it is) rainy and the floor becomes slippery, I (avoid) walking and stay indoors”. A male participant said, “I once slipped and fell accidentally walking on a manhole cover on a rainy day and now I am afraid of it” (PRI9). A female participant (PUB20) noted, “Sometimes on hot days, the sun is as harsh as an oven … The first thing to fear is sunstroke. The second thing is that many older adults have high blood pressure … It is impossible (to walk) on hot days”. Therefore, participants preferred to walk in areas with cover. A male participant (PRI18) said, “We would like to walk indoors or somewhere with cover. It’s more comfortable and the air is better. Now it’s so hot and we walk less”. A female older adult (PUB2) remarked, “We take an umbrella when it rains. It would be much better if there’s (some) cover”.

### 3.4. Policy

#### 3.4.1. Lack of Walking Programs in the Community

The participants reported a general lack of walking interventions organized in the community, except for one-off picnic programs, such as a farm visit. Participants did not know what a walking program involved and misunderstood the concept of a walking program. For example, a female participant (PUB14) stated, “I think there is no need to teach people (how) to walk; that’s what physiotherapists teach you”. A male participant (PUB15) misunderstood walking interventions as “traveling”. When asked if they knew of any walking program in their community, one male participant (PUB19) said that he did not, and a female participant (PRI7) explained that she was not aware of such programs in the community, either because they do not exist or because they are not promoted. In general, more than half of participants believed more walking programs are needed.

#### 3.4.2. Supportive Culture for an Active Lifestyle

A healthy living culture and policy interacts with people’s health benefits at the individual level. Most participants agreed that the overall sporting culture in Hong Kong was positive for older adults’ PA, including walking. A male participant (PUB7) said, “There is a lot of publicity on TV and in the newspapers about doing exercise, otherwise your health will get worse and worse. I want to get fitter and healthier, so I try to do as much exercise as possible”. Moreover, linked to social environment level, an active community culture can increase people’s social interaction in PA such as walking. A female participant (PRI2) said: “Indeed. (The policy) did a lot of sports publicity to enhance the sports culture in the society … Some of my friends who don’t like doing PA have been asked out and lead by their friends to do PA”.

However, in replying to the same question, a female participant (PRI8) lamented, “It is not enough, (especially) compared to (the) USA; places in (the) USA are suitable for walking because houses are isolated and roads are for walking”. Another male participant (PRI3) mentioned: “I think there are not enough (PA/walking) facilities because too many people wanted to join activities”. “Improve the quality of the facilities, so as to prevent the older adults from hurting easily (when they do PA)”, female participant (PUB3). That interacted with walking paths or sidewalks at the physical environment level that government might improve walking-related facilities to enhance older adults’ walking (Figure 1).

Table 2 shows the details of participants and their perceptions toward walking. Health benefits and barriers; social support and social interaction; pedestrian facilities, density, land-use mix, and connectivity were the most significant themes in individual, social, and physical environment, respectively.

## 4. Future Walking Intervention Recommendations

While asking participants about future walking intervention recommendations, they suggested walking in a group, peer-led, intergenerational setting, and professional supervision setting interventions. Adding some interesting and fun activities would also be more attractive to older adults. For instance,

A female participant: “I will go if others joined. I prefer going as a group” (PRI11). Another male participant agreed, “Yes. If center holds walking trips, we can walk as a peer-led group, and chat with each other” (PRI3).

Interestingly, a male participant suggested walking with grandchildren could be more motivating. “Don’t underestimate the young people, for example, at school activity talks, promoting parent-child activities, they will go back and tell their grandparents, in short, give them a chance to come together” (PRI2).

Participants also suggested to have a professional supervisor in the intervention that could teach them knowledge about walking or reducing their fall risk. A female participant: “Teaching us the correct postures of exercises” (PUB3). Another female participant: “Teach us how to walk. Coach us how to walk while maintaining a good balance because balance is … this is better” (PUB4). A male participant: “Oh, the (walking) posture … is quite good for me. It’s because I don’t know what posture is correct although I have exercise for quite a long time” (PUB8). One female participant: “You can show them how to choose comfortable shoes or assistive tools so that those who may not be able to walk easily can walk” (PRI8). Another female participant: “It’s good to have it (professional supervisor). I will join it if there’s one” (PRI18; female).

In addition, people also mentioned having some additional interesting and fun activities could increase their enjoyment in a walking intervention. A female participant: “Then all are walking. Maybe (have) some ball games?” (PRI18). Another female participant: “Of course it’s the walking activity with food to eat” (PRB11). “Some gift or prize”, (PUB10) a female participant added. One male participant mentioned: “So they would take some photos on the bugs and flowers. You’ll realize you have some purpose on walking then walking is not boring anymore … On the way of walking, we can talk or do anything, for example try to learn taking photos” (PRI5).

## 5. Discussion

This qualitative study examined older adults’ perceptions of walking experiences as well as older adults’ suggestions for future walking interventions. The social-ecological model was used to analyze the data that influence the walking behavior of older adults in Hong Kong. The multilevel themes’ influence on older adults’ engagement in walking are presented in Figure 1. The figure illustrates the four levels (individual, social environment, physical environment, policy levels) of influence on older adults’ perceptions on walking. The themes in levels are dynamic and interact.

### 5.1. Individual Level

#### 5.1.1. Health Benefits and Barriers

Factors included being physically active, feeling relaxed and happy, and being frustrated by physical limitations. Most participants mentioned improvements in their psychological health after walking (e.g., happiness and relaxation), and these findings are consistent with a related meta-analysis [9] showing evidence that walking benefits mental health (e.g., reducing psychological stress). This theme, in fact, interacted with the next level, social environment theme, that “social interaction” was one of the factors that influenced their happiness (one of the mentioned health benefits) while walking, as chatting to friends during the walk made them experience enjoyment. Similar to the results of a systematic review [46] that examined the barriers to and motivators of PA for older adults, the current study revealed that improvement in health conditions via walking (e.g., improved blood pressure control) could serve as a facilitator for older adults to walk. However, ill health conditions were also regarded as a barrier to walking [9]. This theme also interacted with social integration at the social environment level. Specifically, some older adults with ill health conditions who walked slowly preferred to walk alone because they felt incompetent, isolated, and unpleasant while walking in group. These results align with the results of other research [20,21,46].

#### 5.1.2. Fall Risk

The most critical individual concern related to walking was fear of falling. Some participants with fall experience(s) feared falling again and lacked the confidence to walk on slippery and uneven roads. This finding echoes the study of Franco and her colleagues [47], that past falls increase older adults’ anxiety regarding walking. Crucially, fear of falling may prevent older adults from undertaking activities (e.g., walking) that may expose them to risky environments such as uneven ground surface.

#### 5.1.3. Perseverance

Some active participants mentioned that their perseverance and determination prompted them to walk every day regardless of poor weather. When facing setbacks in walking, those who employed more effort and determination develop higher self-efficacy [48,49]. This confirms the related research findings that self-efficacy is a facilitator of PA participation [50]. In addition, walking with peers can increase older adults’ perseverance. Peers in a supervisory role can improve a person’s adherence to walking [51], walking self-efficacy, and autonomous motivation [52], which interacts with the social environment level.

#### 5.1.4. Suitable Activity for Older Adults

In line with Yarmohammadi et al. [46] and Zurawik [53], the current study indicated that older adults perceive walking as an economical form of PA because walking is a low-cost activity that requires no equipment or special skills. In addition, walking is flexible and convenient because it can be performed at any time and in any environment, making it a suitable activity for older adults [53].

### 5.2. Social Environment

In this study, social support and social interaction affected older adults’ walking. Receiving family encouragement, being accompanied by friends, and obtaining professional instruction motivated them to walk more; these findings are consistent with those of previous reviews [46,47,54]. This theme interacts with the individual level in that greater social interaction gives greater enjoyment of walking to older adults [54]. Again, social support and interaction also increases older adults’ perseverance because social support and interaction in group walking increases older adults’ PA adherence [51], walking self-efficacy, and autonomous motivation [52]. Next, support from active older adults may improve their inactive peer’s attitude to walking (e.g., perseverance) by setting up a favorable example [50].

### 5.3. Physical and Environmental Factors

#### 5.3.1. Walkability and Accessibility

Related themes include residential density, land-use mix, and connectivity. In contrast to Western countries, where higher residential density might promote walking [55], the high-density residential areas of Hong Kong represent a barrier to walking for older adults. Our result is, in fact, supported by Lu’s research team [56], which reported that lower residential density was positively associated with increased walking in Hong Kong. Participants tended to avoid walking in high-density neighborhoods in Hong Kong. Our finding that land-use mix and connectivity favor older adults’ walking habits were also consistent with the results of previous studies in Hong Kong [28,31]. Higher land-use mix and connectivity are linked to increased walking in older adults in Hong Kong.

#### 5.3.2. Safety

Most participants were concerned about traffic-related safety, such as high traffic volume, reckless driving behaviors, and crossing characteristics, which align with results of a previous review [57]. Notably, no participant mentioned crime-related concerns as a barrier to walking; the low violent crime rate in Hong Kong (0.07 per 1000 population as of 2019; Hong Kong Police Force [HKPF], 2019) and the higher-density environment may increase people’s sense of personal safety when walking [56].

#### 5.3.3. Pedestrian Facilities

Related evidence [12,28,31] suggests that poor pedestrian infrastructure could discourage older adults from walking. Similarly, many of our participants were dissatisfied with existing pedestrian facilities, including insufficient benches and lighting as well as unsatisfactory pavement conditions. These infrastructural shortcomings also relate to the themes of perceived safety and fall risk in individual level because insufficient light and unsatisfactory pavement conditions are considered environmental factors related to older adults’ fall injuries [58], and they should, therefore, be important targets for future town-planning efforts to enhance older adults’ walking.

#### 5.3.4. Aesthetics, Weather, and Other Pedestrian Behaviors

The themes identified in the current study were congruous with related studies that reported green and clean neighborhoods facilitated PA [12,23], because people feel enjoyment when they are exposed to an aesthetic walking trail [59]. In turn, these aesthetics’ factors are related to older adults’ enjoyment of walking. In contrast, unfavorable weather (e.g., rainy and hot days) and poor behaviors by other road users [23] discouraged older adults from walking. In the United States, Dworkin and colleagues [60] observed that precipitation was not significantly associated with older adults’ walking behaviors. However, weather might particularly affect older adults’ walking in Hong Kong because Hong Kong has a humid subtropical climate with very hot (>31 °C) and humid summers. Participants’ experience of these environmental factors influences their feelings and walking behaviors due to “fall risk” and “health benefits and barriers” at the individual level [61].

### 5.4. Policy Level

#### 5.4.1. Lack of Walking Programs

Participants highlighted a lack of walking programs in the community. Kassavou et al. [62] performed a meta-analysis on walking interventions in older adults. Among the 11 included studies, only three walking interventions were conducted in Asia (one in China and two in Japan). This emphasizes the rarity of walking interventions in Asia.

#### 5.4.2. Supportive Culture for an Active Lifestyle

Participants generally stated that sports culture in the community is active and that is in line with government’s initiatives to promote active aging. Examples of these initiatives include the establishment of elder academies to promote lifelong learning (e.g., health education in evergreen learning) in older adults and the Public Transport Fare Concession Scheme for Older Adults. Such programs enhance older adults’ mobility and their awareness of PA (including walking) health benefits (i.e., theme “Health benefits and barriers”) at an individual level. Next, participants perceived an active community culture could increase social integration (at social environment level), for example, sport campaign gathers older adults taking part in sport together. Greater London Authority [63] reported that of all activities, sport has particularly high engagement from a diverse range of social groups.

### 5.5. Future Walking Intervention Recommendations

Based on the results of recommendations for future walking interventions from older adults, older adults believed that professional instruction in a group setting would increase their interest or motivation to join the intervention. That may be due to their perceived self-efficacy level. Kassavou et al. [62] found that group-based walking could promote individual walking efficacy, thereby significantly contributing to higher levels of walking. What is more, some participants suggested to walk with their grandchildren. Intergenerational walking intervention can also be considered. Some studies had suggested that older adults showed higher active PA engagement and well-being from intergenerational intervention [64]. In addition, using professionals in interventions such as physiotherapist, coach, or tutor can increase older adults’ awareness about the health benefits or proper knowledge of walking. All these resulted in better adherence and better perception of one’s physical abilities in future walking intervention [65]. Other than professional supervision, peer-led approaches could be considered, because they are as effective as those led by professionals [66]. Similarly, practical walking recommendations, such as knowledge of proper walking posture, balance enhancement, fall prevention, and shoe selection, are also big concerns for older adults. Next, in line with a meta-analysis about future PA interventions for older adults [67], future researchers or practitioners may include some additional interesting and fun activities such as photo taking during walks to increase older adults’ intention to join walking intervention and help older adults enjoy the walking intervention.

However, as for the environmental factors, such as crowdedness, flat walking path, hot weather, speeding automobiles, and too-short crossing times at traffic lights, these may require expensive measures involving urban construction and policy. In the short run, researchers may consider developing a walking intervention in a temperature-controlled indoor sport complex that has sufficient lighting and a flat walking path. Future researchers or practitioners may take the above walking intervention recommendations as reference and develop future walking interventions that fit their context.

Last, but not least, these synthesized future walking intervention recommendations are, in fact, in line with those significant themes mentioned in the results. For instance, these recommendations echoed themes the “health benefits and barriers” (i.e., having professionals to teach them walking and reducing fall risk at walking), “social support and social interaction” (i.e., peer-led group walking intervention), and pedestrian facilities (i.e., sufficient light provision).

## 6. Study Strengths and Limitations

This is the first theory-based qualitative study to investigate older adults’ perspectives on walking in Hong Kong and their suggestions for future walking interventions. Participants were stratified by sex, type of elderly center, and PA level, which ensured our study’s validity. However, participants were recruited from only four centers and our sampling was slightly different than our expectation (i.e., stratification by activity level) that may have affected our study’s result’s generalizability. Next, since most of the participants were members of the elderly centers, that may have resulted in an over-representation of ‘active’ older adults, as those who attend the elderly center regularly tend to be ‘active’. Future research among hidden ‘inactive’ older people is warranted.

## 7. Conclusions

Our findings provide practical insight for the planning of future walking intervention programs for older adults. For example, researchers and practitioners may consider including injury-prevention training (e.g., fall prevention) to reduce older adults’ fear of falling. We recommend introducing more group-based supervised walking programs for older adults with similar walking competence. The themes related to the physical environment also provide useful information for the government as it endeavors to provide supportive physical walking environments to promote active aging. Particularly, local urban planners or designers may incorporate design considerations (i.e., walkability and accessibility, safety, pedestrian facilities, aesthetics, and other pedestrian behaviors) that may enhance the physical environment and consequently promote walking in older adults in Hong Kong. Future qualitative studies may investigate older adults’ perceptions of walking for different purposes (i.e., recreation and transportation). Future researchers may also consider self-selection and car ownership in their investigations. This is because self-selection (e.g., older adults’ preference for walking) may affect their walking behavior [57], for example, physically active participants would be perceived to have more group-based and social content walking interventions. Moreover, car ownership in Hong Kong is increasing. The number of private cars in Hong Kong increased by 48% from 2008 to 2017 [68]. This may discourage walking in older adults because of higher traffic volume and an increased dependence on cars for commuting. Future policies can be promulgated to encourage green travel (such as public transport subsidy), restrict traffic flow, divide pedestrian and vehicular traffic, and widen roads in order to improve walking environmental quality. Additionally, with the suggestion from older adults, group-based and intergenerational walking intervention with professionals’ assistance will be recommended in the future. Additionally, future walking interventions may include some additional fun activities so as to increase older adults’ intention to join the intervention and help older adults enjoy the experience of walking.

## Figures and Tables

**Figure 1 ijerph-18-07686-f001:**
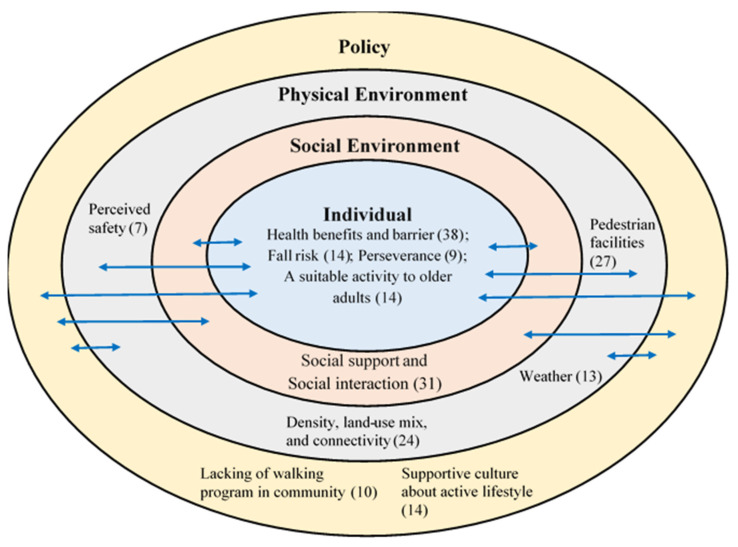
Studied themes in this study. Note: Number in parentheses reflect the number of times that particular theme was mentioned in the interviews.

**Table 1 ijerph-18-07686-t001:** Characteristic of participants.

Characteristic	*N*
Gender	
Female	19
Male	19
Age (years)	
60–65	2
65–70	14
71–80	15
>80	7
PA level	
Active	25
Inactive	13
Nature of Elderly Center	
Public	20
Private	18

**Table 2 ijerph-18-07686-t002:** Interview participant details.

					Individual				Social	Environment					Policy	
Participant	Center	Age	Sex	PA Level	HBB	FR	P	SA	SSSI	DIC	PS	PF	OPB	W	LWP	SC
PUB1	Public	82	F	Inactive	✓		✓	✓		✓	✓					
PUB2	Public	69	F	Active	✓	✓	✓	✓	✓	✓	✓	✓				
PUB3	Public	70	F	Inactive	✓				✓		✓				✓	✓
PUB4	Public	73	M	Active	✓	✓			✓	✓	✓		✓			✓
PUB5	Public	83	M	Active	✓				✓							
PUB6	Public	73	M	Active	✓	✓	✓		✓	✓		✓			✓	
PUB7	Public	83	M	Active	✓			✓	✓							✓
PUB8	Public	60	M	Active	✓			✓	✓		✓			✓		✓
PUB9	Public	72	M	Active	✓				✓	✓		✓		✓		
PUB10	Public	69	F	Active	✓	✓			✓	✓				✓	✓	
PUB11	Public	73	F	Active	✓		✓	✓	✓		✓	✓				✓
PUB12	Public	77	M	Inactive	✓			✓	✓		✓	✓				
PUB13	Public	76	F	Active	✓	✓	✓		✓	✓		✓		✓		✓
PUB14	Public	78	F	Active	✓	✓			✓	✓		✓		✓	✓	
PUB15	Public	69	M	Active	✓				✓	✓					✓	
PUB16	Public	72	F	Inactive	✓	✓			✓							
PUB17	Public	81	F	Active	✓		✓		✓	✓		✓				
PUB18	Public	84	M	Active	✓				✓	✓		✓				
PUB19	Public	70	F	Active	✓				✓			✓			✓	
PUB20	Public	68	M	Inactive	✓	✓		✓				✓		✓		
PRI1	Private	66	F	Active	✓		✓		✓	✓		✓				
PRI2	Private	71	M	Inactive	✓	✓				✓		✓		✓		✓
PRI3	Private	73	M	Active	✓			✓		✓		✓				✓
PRI4	Private	68	M	Inactive	✓			✓	✓	✓		✓		✓		
PRI5	Private	65	M	Active	✓		✓		✓			✓				
PRI6	Private	70	F	Inactive	✓				✓	✓		✓			✓	
PRI7	Private	68	M	Active	✓			✓	✓	✓		✓		✓	✓	
PRI8	Private	67	F	Inactive	✓					✓	✓	✓			✓	✓
PRI9	Private	72	M	Active	✓	✓	✓			✓		✓				✓
PRI10	Private	69	F	Active	✓			✓	✓			✓	✓	✓		
PRI11	Private	71	F	Active	✓				✓					✓		✓
PRI12	Private	79	F	Active	✓	✓			✓	✓		✓		✓		✓
PRI13	Private	91	F	Inactive	✓	✓		✓	✓							
PRI14	Private	86	M	Inactive	✓	✓				✓		✓				
PRI15	Private	66	F	Inactive	✓				✓			✓				
PRI16	Private	78	F	Inactive	✓			✓	✓	✓		✓			✓	
PRI17	Private	53	M	Active	✓				✓	✓		✓		✓		✓
PRI18	Private	72	M	Active	✓	✓		✓	✓	✓		✓	✓			✓

Note: HBB: health benefits and barriers; FR: fall risk; P: perseverance; SA: a suitable activity for older adults; SSSI: social support and social interaction; DIC: density, land-use mix, and connectivity; PS: perceived safety; PF: pedestrian facilities; OPB: other pedestrian behaviors; W: weather; LWP: lack of walking programs in the community; SC: supportive culture for an active lifestyle.

## Data Availability

According to the research ethics of Hong Kong Baptist University, all the personal data of the participants are kept conferential in any publication of result of this study. The material will be maintained for up to 3 years.

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
