# Peer review of "Older Adults’ Perceptions toward Walking: A Qualitative Study Using a Social-Ecological Model"

_ijerph, 2021, doi:10.3390/ijerph18147686_

Round 1

Reviewer 1 Report

This paper is about an important and relevant topic, largely aligned with existing research on this topic, and makes a small contribution to our understanding of older people's perceptions of walking in high-density cities. 

It requires a significant revision to merit publication. The following major weaknesses should be addressed: 

  1. The authors need to explain why the research gaps are important to fill. a) What do qualitative studies add that quantitative studies do not? Why is it important to do more qualitative research? b) Why do studies in Western countries not apply to places like Hong Kong? (Also, consider generalizing this to large mega-cities in Asia--not just Hong Kong). What environmental factors make the research done in Western countries not applicable to Eastern countries, especially large mega cities in Asia? c) What does the socio-ecological model add that other theory-based studies fail to address?
  2. I would like to see a shorter introduction that deals with the main research questions and gaps, and then a more detailed literature review that goes into more detail. Tell me what the general themes are in the existing research--this will help me understand how your own research is enhancing what has already been done. 
  3. A weakness of qualitative research is that it often feels very anecdotal. It's not clear how many of the participants commented on each of the different themes. That leads me to wonder whether just one or two people addressed health benefits (for instance) or the majority of the participants. Please review the data and provide numbers that account for how many participants expressed the different view points. (e.g.,  "15 of the 38 participants regarded improved health as a motivation for walking..."). 
  4. It was not clear to me that participants wanted a walking program or felt they were beneficial, yet that seems to be one of your major policy recommendations. Please explain how the results justified your conclusion. The results seem to suggest that there were no walking interventions people were aware of, but it wasn't clear to me that people wanted more walking programs or felt like they could benefit from them. 
  5. Please address more of the limitations of the studies. How does selecting participants from senior centers limit the kinds of participants you selected? How might self-selection bias influence the results (it seems that the majority of your participants were physically active)? Did you control for socio-economic status? Location? It's striking that none of the participants indicated that safety was a concern, which leads me to believe that the participants came from more well-to-do areas of the city. 
  6. Although you asked how many people were physically active (150 minutes of PA per week), it was not clear to me how many people regularly walked for exercise--or to get around. Is this a question covered by your protocol? It seems important to have this baseline.

Other concerns that address the overall writing quality include:

  1. Many problems with the punctuation (placement of periods, commas, elipses, quotation marks, etc.). Please refer to a good style guide.
  2. An introductory paragraph that has nothing to do with the overall theme of the paper--yes, there's an aging population. But this paper is about physical activity and older adults. I recommend deleting the first paragraph.
  3. Long paragraphs that should be split up to emphasize the main ideas and to make it easier for the reader to understand (see, for instance, page 2 and page 5). 
  4. Problems with prepositions and articles that a native speaker should be able to fix. I recommend hiring a good editor to address these problems. 
  5. Longer quotes should be indented. 

I hope these overall suggestions will make your paper stronger. Best of luck! 

Reviewer 2 Report

This is a potentially interesting contribution to our understanding of barriers for the elderly to walk in our communities. However, as presented, the results and discussion really do not advance our understanding. There is no synthesis with existing research in Hong Kong or internationally. This paper could really do more work in emphasizing the key barriers to walking and the key potential interventions. As it is, it feels like half a manuscript and the hard work of interpreting the results and making it useful to other researchers and the community in Hong Kong has not be done. Finally, add a conclusion section. The manuscript sort of sputters off after listing many areas for future research. Make a strong conclusion.

There are some minor typographical errors, like spaces or periods in the wrong places. Also, some of the in-text references are not of the correct style.

In section 2.1 Participants, how many participants were contacted to participate? How were they recruited? How were the elderly centers selected? "purposive sampling" is vague. State explicitly how you drew your sample, the sample frame, and the sampling methods.

For the sample size, "meeting the sample guidelines" what is the effect size that you determined to meet guidelines? For which question or measurement did you determine the sample size?

Table A1. Provide the percentages as well.

Why was the percent of your sample meeting PA so high (58% vs 30% in the population)? Was this purposive? Please discuss.

Table A2. is unnecessary, remove.

Section 2.3 Procedures. Please move the details about recruitment up to the 2.1 Participants section. Procedures should be about the interview process, not the recruitment.

Were any respondents ineligible? Please state numbers and reasons.

Results

Where are the results of the Timed Up and Go Test and mental test? How many participants were excluded? Why would these exclude participants? Do participants not walk because of these measures? It does not seem like these would be disqualifying... It is unclear as written.

The reporting of the thematic coding of results could be better. The authors review a lot of categories, but there is no summary of similar response frequency within categories. Therefore, it is difficult to see within categories, what is the most commonly reported concern.

Also, there is no ranking of the responses, so it is not possible to determine what the most important barriers are to walking between categories. Is it the weather (something we cannot change) or is it speeding automobiles (something we can change)? The results are presented all as equivalent, maybe because that is how the questions were asked? Please re-think the presentation of the results.

Anyways, this severely limits how much we have (potentially) learned about this population and does not really inform or prioritize targets for intervention. Given all the barriers to walking, where do you start? Would policy interventions or social supports really overcome the stated barriers of crowdedness, hot weather, speeding automobiles, too short crossing times at traffic lights, etc.? The evidence presented does not point to one key intervention, but multiple, expensive, and likely infeasible interventions. Please add this to your discussion and limitations section.

Presenting all of the barriers and facilitators as equivalent really does not add to our understanding. Try to tease apart the responses to see what appears to be most important and most addressable. Maybe, add a table that ranks the potential interventions to the stated barriers on practicality and cost? That would really add something.

Reviewer 3 Report

Dear Editors,

Have a nice day!

This paper investigates the mindset of elderly people aged 60 or above from different elderly centers that how they think about their walking experience like what factors push or pull them for going out for a walk. It explored individual, social, and physical attributes that influence an individual’s intentions about walking. The strengths of this paper include innovative work using socio-ecological model to explore the perception of older adults and the method used for this investigation is rigorous. The weaknesses include the organisation and presentation of work. Therefore, I suggest the following revisions:

TITLE: The title of this study is fine and representative to what authors have done in this paper.

ABSTRACT:

  1. methods’ lines are about the sample and tool for the data collection, not about the method. Before this line, authors should add a line that they used thematic coding method for this study as mentioned in 2.4.
  2. The keywords should be different from the keywords used in the title, avoid repetition. It might be like: active aging; physical activity; thematic coding; Hong Kong etc.

INTRODUCTION:

  1. Evidence of the research gap is lacking. Fourth paragraph of the introduction is about the research gap where the authors mentioned similar studies in different countries and claim that such studies are lacking in higher population density cities such as Hong Kong. This gap at the end of the fourth paragraph of the introduction better come with some citations to strengthen the base of this study.
  2. In the end of the introduction, authors should foreshadow the rest chapters of their paper by adding few lines that what readers should expect in the rest parts of this paper.
  3. The authors have used two different styles to cite a reference. The fourth line of the fourth paragraph and many other locations throughout the paper double citation of the same reference with two different styles should be removed.

METHODS:

  1. The authors might also be interested in extending the columns of Table A2 in the following way:

Participants and their perceptions (coded)

ID No.

Center

Age

Sex

PA level

Individual  

Social

Physical

3.1 *1HBB

3.2

*2FR

3.3

3.4

PUB01

Public

82

F

Inactive

PUB02

Public

69

F

Active

*1 HBB for health benefits and barriers , *2 FR for fall risk

The authors can expand the table by adding more sub-headings under social and physical column like the example given under the individual column. Hence, the dark dots can be used to indicate participants' perceptions touching the areas under individual, social, or physical category.  

RESULTS:

  1. A theme is an outcome of coding and if authors could elaborate a bit more about the journey from coding to a theme would increase the quality of presentation. For example, 3.1.1. Health Benefits and Barriers is a theme generated out of a coding process where ‘happiness’ is a frequent response coded and converted under the 3.1.1. theme as it is taken as participants’ psychological well-being in line 3 of 3.1.1. Therefore, a table under each section of individual, social, physical describing the original responses of participants in one column and other one about the theme will help readers know the process of conversion (into theme) in a better way.  
  2. There should be a conclusion part in the end of the paper. It might be like CONCLUSION: study strengths and limitations

The review draft is without line numbers. It was difficult to pinpoint the exact location where the revision is required. Any future submissions should be according to the given format on the journal website.

Reviewer 4 Report

The article presented is of great interest and relevance. It is worth lending to how a meaningful activity such as walking can be used therapeutically with older people.

Here are a number of recommendations that could enhance the great article presented:
- The last paragraph of the introduction should be devoted to explaining in detail the aim of the paper. Right now it is ambiguous and the authors have incorporated very different issues in that paragraph.

- It would be interesting if the authors could explain in more depth how they recruited those particular participants. It is also unclear how many residential facilities the elderly participating in the study belong to.

- Similarly, it would be relevant to mention the ethical issues involved in the study conducted.

- It is necessary to include a final section with the conclusions of the study. 

Round 2

Reviewer 2 Report

The goals and significance of this study are still unclear. The synthesis in the Discussion section in inadequate. The authors did not revise this manuscript to make a convincing case to be published.

Author Response

Dear reviewer,

Please find attached document. 

Best,

Karena

Reviewer 3 Report

Dear Editors, 

Have a nice day. 

The authors have sufficiently improved the quality of their work according to the given suggestions.  

Author Response

Dear reviewer, 

Thanks for your suggestions, which has helped us a lots.

Best,

Ms. Ou